# PhotoModPlus: A web server for photosynthetic protein prediction from genome neighborhood features

Apiwat Sangphukieo[1,2], Teeraphan Laomettachit[1], Marasri Ruengjitchatchawalya[1,3,4] *

1 Bioinformatics and Systems Biology Program, School of Bioresources and Technology, King Mongkut's University of Technology Thonburi (KMUTT), Bang Khun Thian, Bangkok, Thailand, 2 School of Information Technology, KMUTT, Thung Khru, Bangkok, Thailand, 3 Biotechnology Program, School of Bioresources and Technology, KMUTT, Bang Khun Thian, Bangkok, Thailand, 4 Algal Biotechnology Research Group, Pilot Plant Development and Training Institute, KMUTT, Bang Khun Thian, Bangkok, Thailand

* marasri.rue@kmutt.ac.th

**Data Availability Statement:** All relevant data are within the manuscript and its Supporting Information files.

## Abstract

A new web server called PhotoModPlus is presented as a platform for predicting photosynthetic proteins via genome neighborhood networks (GNN) and genome neighborhood-based machine learning. GNN enables users to visualize the overview of the conserved neighboring genes from multiple photosynthetic prokaryotic genomes and provides functional guidance on the query input. In the platform, we also present a new machine learning model utilizing genome neighborhood features for predicting photosynthesis-specific functions based on 24 prokaryotic photosynthesis-related GO terms, namely PhotoModGO. The new model performed better than the sequence-based approaches with an F1 measure of 0.872, based on nested five-fold cross-validation. Finally, we demonstrated the applications of the webserver and the new model in the identification of novel photosynthetic proteins. The server is user-friendly, compatible with all devices, and available at bicep.kmutt.ac.th/photomod.

## Introduction

Photosynthesis is an important, complex biological process by which solar energy is converted into biochemical energy by some diverse organisms, including plants, algae, cyanobacteria, phytoplankton, and photosynthetic bacteria, for the production of biological compounds. However, there is still limited information about the complexity of the photosynthetic system, and many genes and protein functions are yet to be explored. Discovering new genes/proteins in the photosynthetic process is substantially important, as it may point the way to improve photosynthesis efficiency [1–4] and its applications [5–8].

Considering the efficient pipeline used in the identification of novel photosynthetic proteins, it is clear that computational approaches are a crucial screening step preceding in-depth studies by experimental approaches [9–11]. Earlier attempts at computational discovery had

**Funding:** This work was partly supported by Petchra Pra JomKlao Doctoral Scholarship (No: 13/2558) from King Mongkut's University of Technology Thonburi and a research grant (NRMJ: 2559A30602134#60000108) from the National Research Council of Thailand (http://en.nrct.go.th). The funders had no role in the study design, data collection and analysis, decision to publish, and preparation of the manuscript. There was no additional external funding received for this study.

**Competing interests:** The authors declare that they have no competing interests.

been based on sequence similarity search, which is trustable at high sequence similarity, but its accuracy drops as the similarity declines. It was shown that the classification of photosynthetic proteins suffers from the diversity of photosystems, which can cause a high false-positive rate of up to 70% [12]. Moreover, it is inapplicable in the absence of similar sequences in databases. Therefore, techniques using different sources of the feature have been developed to overcome the challenge posed by classical sequence similarity-based methods [13]. Among them is SCMPSP [14], a genetic algorithm model that uses the dipeptide property and amino acid composition to classify photosynthetic proteins. SCMPSP performs better than the sequence similarity search method but is not significantly better than other machine learning models, indicating a limitation of the sequence-based feature. Therefore, a model called PhotoMod, which employs the genome neighborhood network (GNN) as a feature, was developed [15]. PhotoMod combines protein clustering, genome neighborhood conservation scoring, and the random forest (RF) algorithm to classify photosynthetic proteins. It was shown that the genome neighborhood-based model outperformed sequence-based approaches and showed the potential to predict novel photosynthetic proteins. However, this method requires a tedious installation process, which might prove obstructive and demanding, if not discouraging, for some users. Moreover, it can only classify proteins into two categories: photosynthetic and nonphotosynthetic proteins, and the remaining big task is to investigate functional subclasses in photosynthesis by laboratory tests.

It has long been known that photosynthesis consists of several subsystems, including photosystem I, photosystem II, electron transport system, ATP synthase, NADH dehydrogenase, light-harvesting complex, carbondioxide fixation, and many assembly factors and regulators. However, many proteins in these systems, for example, proteins in ATP synthase and NADH dehydrogenase, can be homologously observed in nonphotosynthetic organisms. Thus, to systematically define the group of photosynthetic proteins, Ashkenazi et al. [12] created a list of functional terms that are unique to photosynthesis based on the gene ontology system to identify photosynthetic proteins.

In general, existing protein function prediction methods might be applied, although they were not specifically developed to predict photosynthesis subclasses. For example, the SVMprot model, which uses a support vector machine (SVM) with physicochemical features of protein sequences such as amino acid composition, hydrophobicity, polarity, polarizability, and charge, was used to predict 192 functional classes, including three photosynthesis-specific subclasses [16, 17]. DeepGOPlus, a sequence-based method, which combines sequence similarity-based predictions with a deep convolutional neural network, was used for large-scale protein function prediction, including 5,210 functional classes and ten photosynthesis-specific subclasses [18]. Nevertheless, these models are less specific to photosynthetic functions, covering only ten of the photosynthesis-specific classes [12]. Most importantly, sequence-based models tend to perform worse in biological process (BP), which is the main category (in gene ontology resource) of the photosynthesis subclasses [19, 20]. It has also been suggested that ensemble methods combining data from different sources could improve prediction accuracy [20]. Therefore, we hypothesize that the combination of data from sequence features and genome neighborhoods can improve prediction accuracy.

In this study, we developed a new web server, namely PhotoModPlus, to serve users for the prediction of photosynthesis genes/proteins and their functional subclasses. The web server (Fig 1) contains three main applications. The first application, PhotoMod, is our previously developed machine learning model that serves for photosynthetic protein classification [15]. The second, PhotoModGO, is a new machine learning model utilizing multi-label learning and genome neighborhood profile as a feature to predict 24 sub-functional classes of a photosynthetic function. We demonstrated that our new model performs better than two sequence-

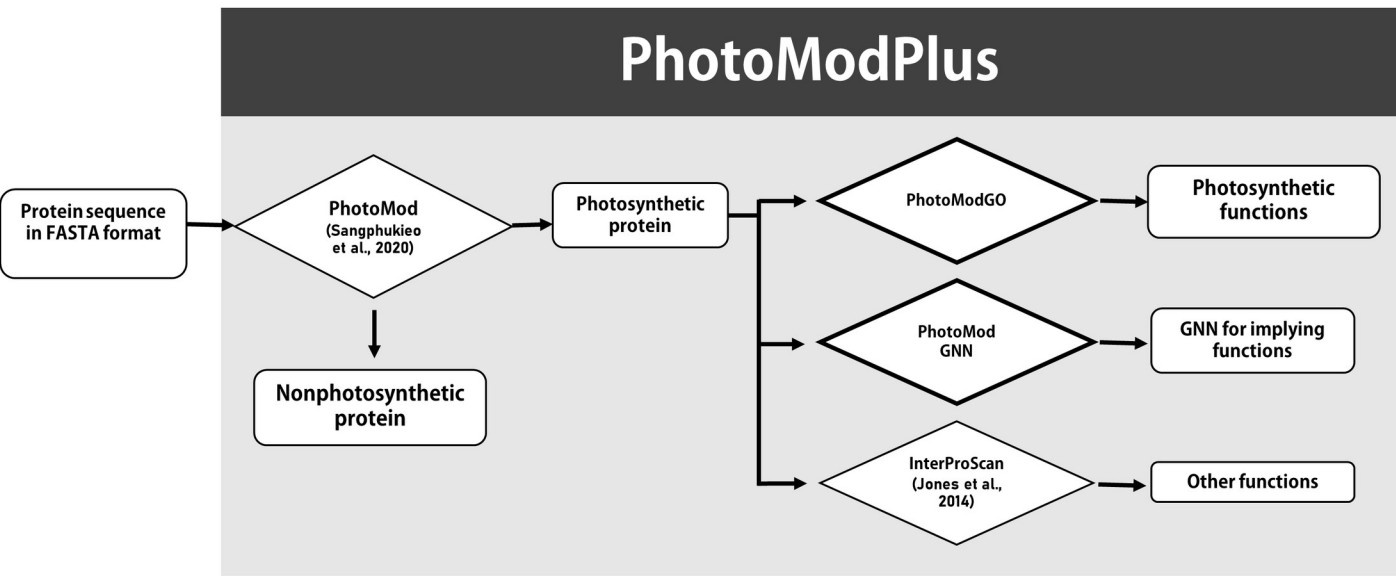

**Fig 1. Workflow of the PhotoModPlus web server for the identification of photosynthetic proteins and related functions.**

based methods, BLAST [21] and DeepGOPlus [18], investigated by nested cross-validation evaluation. The new model serves to explore functional subclasses of photosynthetic proteins, which might be retrieved from the candidates of the first application or other sources. The last, PhotoModGNN, is the application for genome neighborhood network (GNN) generation and visualization, which allows users to observe genome neighborhood patterns and explore photosynthetic functions by network analysis. Our PhotoModPlus provides an easy-to-use web interface for input submission and a convenient interpretable output for photosynthetic protein discovery.

## Method

### Dataset collection

Photosynthetic protein datasets used in PhotoModPlus comprised two types: i) known photosynthetic protein datasets for training and testing the model, and ii) novel photosynthetic protein datasets.

**Training and test datasets.** The photosynthetic protein dataset was retrieved from the UniprotKB database. 15,191 protein sequences consisting of at least one of 61 photosynthesis-specific GO terms identified by Ashkenazi et al. [12] were included in the dataset. To avoid incomplete gene neighborhood identification, only photosynthetic proteins from 154 photosynthetic prokaryotes with complete genomes were included in the dataset, as previously reported [15]. To reduce sequence redundancy, we used the USEARCH analytical tool [22] to cluster similar sequences (sequence identity $\leq$ 50% as a diverse dataset and $\leq$ 70% as an easy dataset), using the command:

$$usearch - cluster\_fast[input] - id[identity] - centroids[output],$$

where the input file is in FASTA format, identity is the percent sequence identity cutoff for the cluster, and output is the selected representative sequence in FASTA format. The number of sequences in the dataset was reduced to 369 sequences with identity $\leq$ 50% and 1,021

sequences with identity ≤ 70%, and these were applied for model development and model comparison.

**Novel photosynthetic protein dataset.** A set of novel proteins collected from the literature was used to evaluate the model performance. Proteins that had never been deposited in the Uniprot database by September 2016 were considered novel as our model was trained using data retrieved then. BLAST was used to check their availability in the database with the criteria: sequence identity ≥ 50% and query and subject coverage ≥ 70%. The lack of sequence homolog makes them ideal for testing the feasibility of our model to facilitate functional annotation of novel proteins.

## PhotoModGO development

The PhotoModGO was newly developed using a multi-label classification approach, which allows each data point to be assigned to more than one functional class at the same time, to classify photosynthesis subclasses of photosynthetic proteins based on the genome neighborhood feature. The data preparation and preprocessing of PhotoModGO are similar to Photo-Mod procedures, allowing a connection between the two approaches, whereby the output from PhotoMod can be used as input data in PhotoModGO. In the pipeline, the PhotoModGO shares data preprocessing steps, including genome neighborhood calling, Phylo score calculation, and Phylo score normalization with PhotoMod. The photosynthetic subclasses were selected from the list of 61 photosynthesis-specific GO terms [12], which are child terms of photosynthesis (GO:0015979) at every level and are not linked to other functions in the BP category. Therefore, they can be considered as a part of the same tree of the photosynthesis term. Among these, only 23 GO terms belong to photosynthetic prokaryotes. In addition, we manually added one term of phycobilisome (GO:0030089), which is a light-harvesting protein complex in cyanobacteria and red alga. Although this GO term is not the child term of photosynthesis, it can be considered as marginally connected to photosynthesis through a photosynthetic membrane (GO:0034357). The description of all 24 selected GO terms is shown in S1 Table.

The process started with the retrieval of the photosynthetic protein dataset with photosynthesis-specific GO terms, followed by the removal of redundant data as described in the dataset collection (Fig 2). Then, the loci of those photosynthetic protein-coding genes were determined and used for calling the gene neighborhood. The genes were considered neighbors if they were within 250 bp on the same strand. Two gene clusters were merged into the same neighborhood gene cluster if they were in a range of 200–1000 bp in the divergent direction, following the operon interaction concept as demonstrated in S1 Fig. The homologous relationship between protein sequences was determined by the protein clustering method with three stringent criteria: 1E-10, 1E-50, and 1E-100, according to a previous study [15]. The genome neighborhood conservation scores (Phylo scores) were calculated based on the phylogenetic tree of organisms that conserve those gene neighborhoods, as described previously [15]. The quantile cutoff points were determined and used for converting the raw Phylo scores to simple numeric forms (i.e., 0, 1, 2, and 3). Hereafter, we call the list of gene neighbors and the Phylo score of the query gene as a genome neighborhood profile. The matrix of the query genes and their genome neighborhood profiles was built and concatenated with the binary matrix of GO term annotations corresponding to the query genes. This processed matrix was converted to ARFF format, which is a standard format for machine learning software. Note that we keep using a matrix form to display the data in Fig 2 as it is easier to understand.

The next steps include the process for building the machine learning model: i) selection of the optimal feature set and ii) training of the model. Due to a large number of features in the

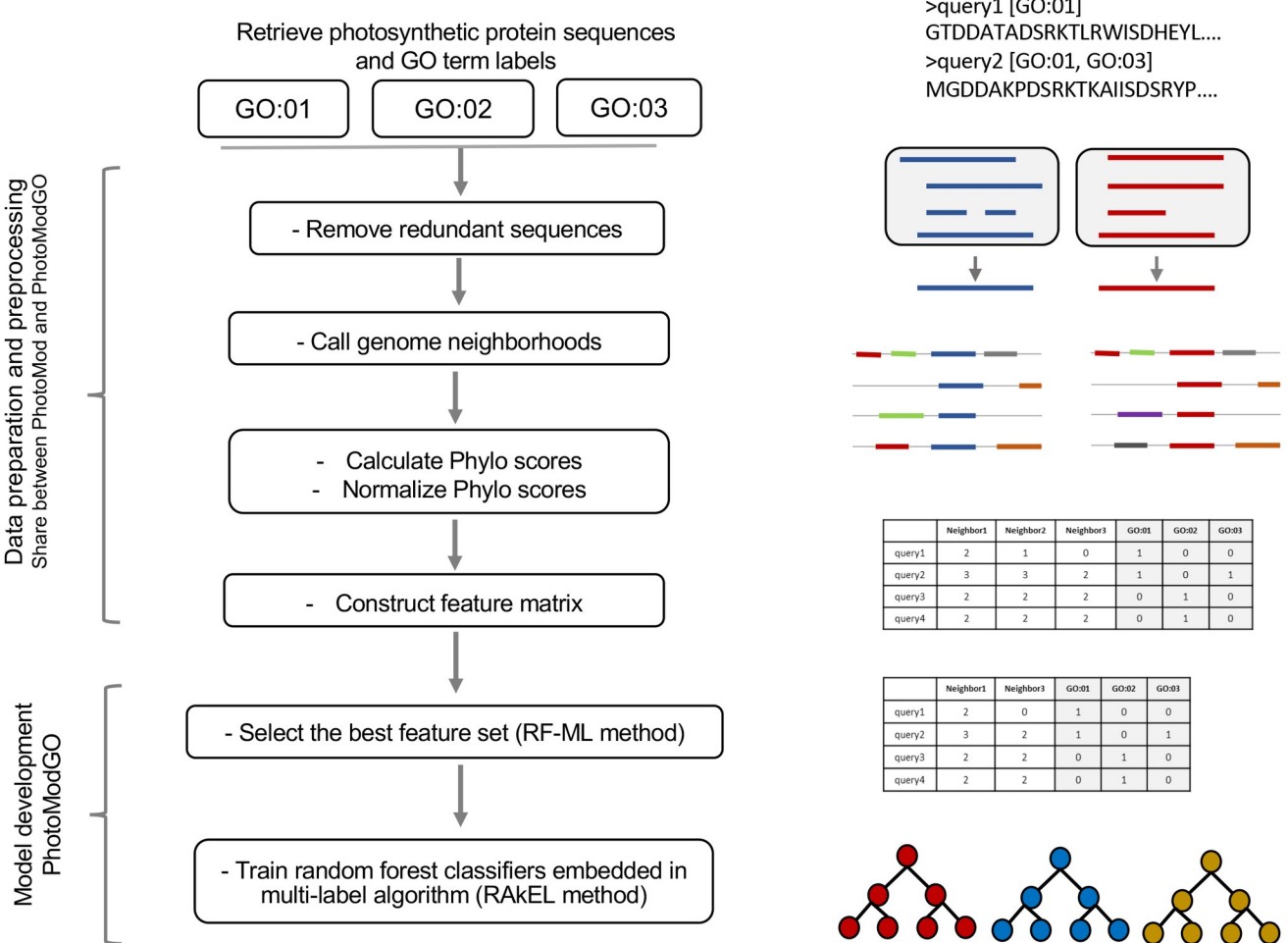

**Fig 2. Overview of PhotoModGO development.** As described earlier, the multi-label model of PhotoModGO was developed by employing feature selection using the RF-ML method before training the random forest classifier embedded in the RAkEL (RAndom k labELsets) algorithm. The right panel shows the graphical representation for each step.

genome neighborhood profile, we removed redundant and irrelevant features in the first step to increase learning speed and model performance. We employed the multi-label feature selection method, RF-ML [23], which is the extended version of the single-label feature selection method, Relief. The RF-ML method was developed using the dissimilarity function between multi-labels to determine the optimal set of features. That is, it considers the relationship between labels, making the prediction performance of the model better. We modified the original RF-ML script to obtain the top-N features ranked by the dissimilarity score, where N is the number of features in the feature subset. The N number was fine-tuned during the model development process.

The next step was the building of the machine learning model, during which we employed a problem transformation method to transform the multi-label dataset into a multi-class dataset, allowing usual learning algorithms to learn. We initially selected three different multi-label transformation methods for model development, and the method with the best performance was selected as the final model. First, we explored the Binary Relevance (BR) [24] algorithm, a classical approach in multi-label classification, which constructs a binary classifier for each label with the final prediction determined by aggregating the classification results from all classifiers. The

problem with this method is that it ignores label relationships, resulting in poor performance when applied to real-world datasets. Next, we explored the Label Powerset (LP) method [25], which takes label dependency into account by constructing binary classifiers from all possible sets of labels. Therefore, it usually suffers from worst-case time complexity when the size of the label set is large. Finally, we used RAndom k-labELsets (RAkEL) [26], an ensemble of LP classifiers, each of which is built from a different random label subset of size k. Therefore, its classification performance depends on the random strategy of the label subset. We applied these transformation methods with the random forest algorithm as a base binary classifier, as it performed the best in our previous study [15]. We developed the classification model using the MEKA framework [27]. Three parameters: the numbers of trees, random features used in the tree in the random forest, and selected features from the RF-ML method, were tuned to optimize the model.

## PhotoModGNN development

The PhotoModGNN is portable, informative, and easy to interpret. As demonstrated in S1C Fig, the hexagonal node represents the query, while the circular node represents its genome neighbors. The labeled text in each node represents a protein cluster ID. The size of the genome neighbor node varies according to its conservation score (Phylo score). There are three built-in protein clustering cutoffs (E-value: 1E-10, 1E-50, and 1E-100) available at the top of the network to define the level of homologous relationship between proteins. These pre-calculations allow users to immediately and dynamically adjust the network according to the stringency. The network is visualized via the Cytoscape.js [28] plugin. Users can click on the node to observe the most enriched functions in each protein cluster in the network. We also provided a link from the node to the hub of protein databases (InterPro), allowing users to gain more insight into protein functions. The output page shows the list of GO terms that are statistically enriched among the group of genome neighborhoods. Users can use this tool for functional guidance, operon prediction, and protein evolution analysis by observing the genome neighborhood patterns. The standalone version of this tool is freely available at https://github.com/asangphukieo/PhotoModGNN.

## PhotoModPlus web server implementation and demonstrations

The web server was developed using Flask framework version 1.0.2 and a Python script and implemented using Apache HTTP web server version 2.4.18. In addition to the previous binary classification model 'PhotoMod' [15], which combines protein clustering, genome neighborhood extraction and scoring, and random forest algorithms to classify photosynthetic proteins, PhotoModPlus includes the development of 'PhotoModGO' and 'PhotoModGNN' in the pipeline for function categorization of photosynthetic proteins.

The workflow of PhotoModPlus is shown in Fig 1. Users can submit single or multiple protein sequences in FASTA format directly as input. Then, the homologs of the input are identified, and the genome neighborhood profiles are automatically generated. The input sequences with genome neighborhood profiles are delivered to PhotoMod [15] for classifying the photosynthetic proteins. Then, users can choose the candidate photosynthetic proteins for the next steps. PhotoModGO can be used to predict photosynthesis-related functions of photosynthetic proteins, PhotoModGNN can be used to create and visualize genome neighborhood network of photosynthetic proteins for inferring their functions, and InterProScan [29] can be used to identify other functions via a protein motif search. Users can use default parameters to submit query sequences, and the link to the output page is sent to the user by email. The web server is available at bicep.kmutt.ac.th/photomod.

**Photosynthetic function prediction by PhotoModGO.**   We demonstrated the application of PhotoModGO by predicting the function of a photosynthetic protein, sll0272. After the

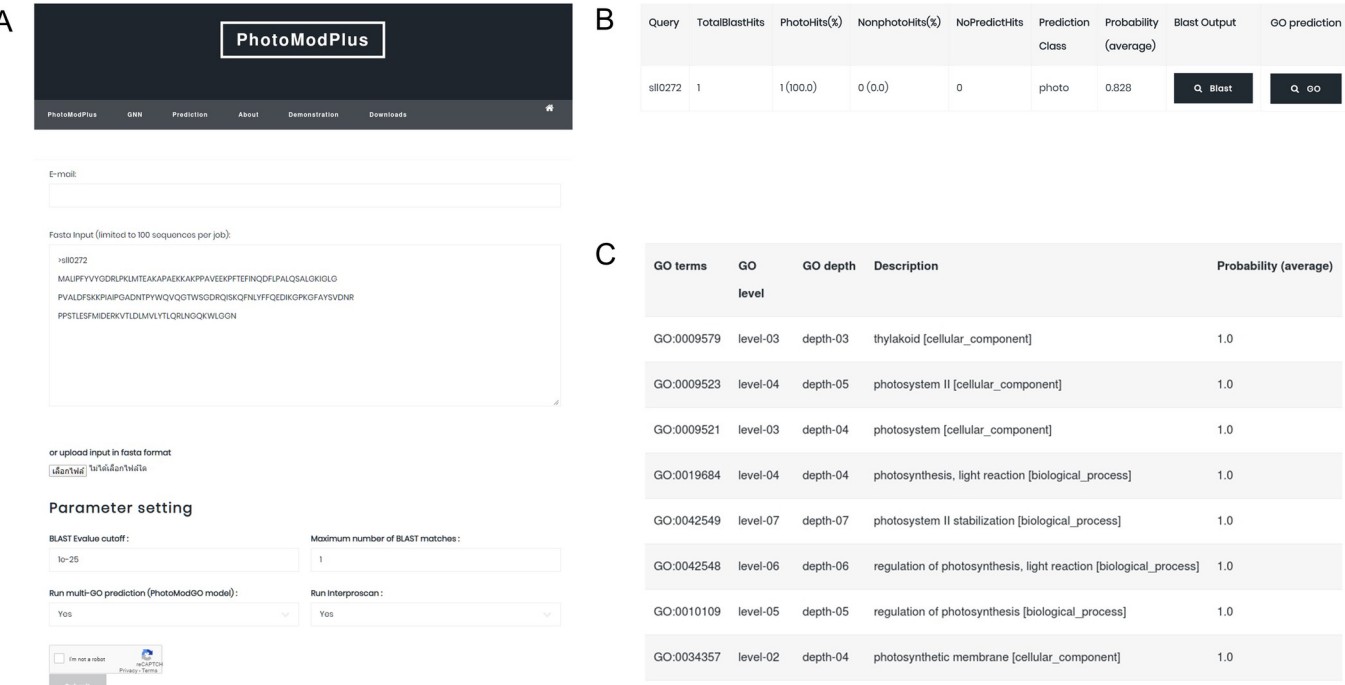

**Fig 3. Function prediction of photosynthetic protein sll0272 using PhotoModGO in the PhotoModPlus web server.** (A) The submission page of the PhotoModPlus web server. Users can submit the input sequence in FASTA format and modify BLAST parameters for matching the query to the sequences in the database. (B) The prediction output from the PhotoMod indicating a high chance of protein sll0272 being involved in photosynthesis (C) The PhotoModGO prediction output indicating the involvement of sll0272 with the photosystem II and regulation of photosynthesis.

homologs of the input are identified with the E-value cutoff of 1E-25 and the maximum number of BLAST-matches limited to 1 (Fig 3A), the genome neighborhood profile is automatically generated. The prediction output from the binary classification model (PhotoMod) indicates that sll0272 has a high chance (83%) of being a photosynthetic protein (Fig 3B), whereas the output from the multi-label classification model (PhotoModGO) indicates the association of sll0272 with the photosystem II and regulation of photosynthesis (Fig 3C).

**Functional guidance by PhotoModGNN.** We demonstrated the application of Photo-ModGNN using the sll0272 protein sequence as an input. The submitted sequences were searched by BLAST against our protein sequence database, which contains sequences collected from photosynthetic organisms only. The GNN (Fig 4A) shows that sll0272 is conserved with 11 genome neighborhoods and moderate Phylo scores in the range of 2.37 to 3.96. The distribution of the Phylo scores collected from the entire dataset is shown in S2 Fig. The list of enriched GO terms (Fig 4B) among the group of genome neighborhoods indicated that the query might be involved in a photosynthesis-related function and nucleic acid-related activity.

## Baseline comparison methods and evaluation

To evaluate the performance of our newly developed model, PhotoModGO, we selected the recently developed sequence-based machine learning model, DeepGOPlus, and standard sequence similarity method, BLAST, for comparison. Nested fold cross-validation and F1$_{max}$ score were used to evaluate the prediction performance.

**DeepGOPlus.** The DeepGOPlus was retrieved from https://github.com/bio-ontology-research-group/deepgoplus.git. We retrained this model using our photosynthetic protein

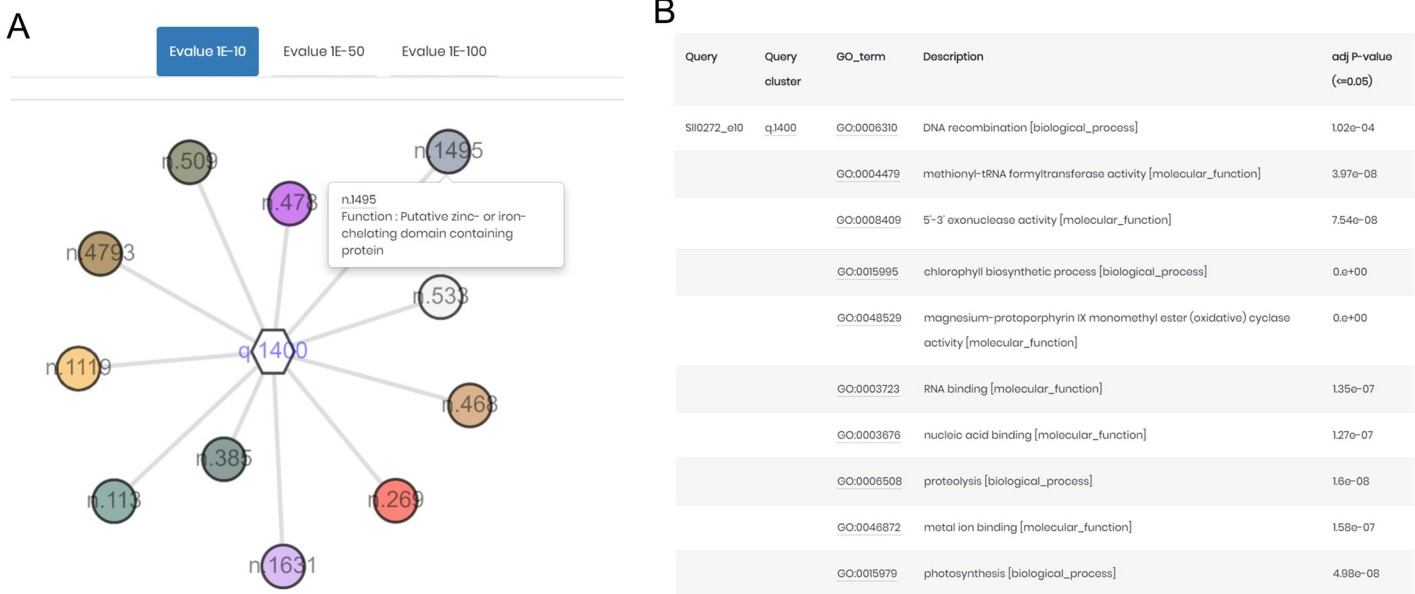

**Fig 4. Example of PhotoModGNN output for sll0272 from the PhotoModPlus web server.** (A) The hexagonal node represents the sll0272 protein cluster, while the circular node represents the protein cluster of the neighboring gene. The size of the node is calculated according to the Phylo score (genome neighborhood conservation score), while the number label indicates a protein cluster ID. The GNN is displayed with an E-value of 1E-10. (B) The list of enriched GO terms among the group of the genome neighborhoods.

dataset. Due to the high computational requirement of DeepGOPlus, we selected two optimal parameter sets, fine-tuned from the original paper, for use in this study.

**BLAST.** We also used a sequence similarity method based on Diamond BLAST as a baseline method for comparison. The training set was used as a database, and the test set as a query. Annotation from the training dataset was transferred to a similar sequence in the test set if it passed the E-value criteria, which ranged from 20 to 1E-30. The percent sequence identity was used as a probability score. The diamond BLAST command is shown below.

$$diamondblastp - d[training\_data] - q[test\_data] - e[E-value] - -outfmt\ 6\ qseqidsseqidpident > [output]$$

**Evaluation metric.** To evaluate the prediction performance, we used simple and easily interpretable metrics, $F1_{max}$ [30], as recommended by the Critical Assessment of Functional Annotation (CAFA) [30]. The F1 score is the harmonic mean of precision and recall, and $F1_{max}$ is the maximum overall F1 score classification threshold of each model.

$$pr_i(t) = \frac{\sum_f I(f \in P_i(t) \wedge f \in T_i)}{\sum_f I(f \in P_i(t))} \tag{1}$$

$$rc_i(t) = \frac{\sum_f I(f \in P_i(t) \wedge f \in T_i)}{\sum_f I(f \in T_i)} \tag{2}$$

$$AvgPr(t) = \frac{1}{m(t)} \cdot \sum_{i=1}^{m(t)} pr_i(t) \tag{3}$$

$$AvgRc(t) = \frac{1}{n} \cdot \sum_{i=1}^{n} rc_i(t) \tag{4}$$

$$F1_{max} = \max_{t} \left\{ \frac{2 \cdot AvgPr(t) \cdot AvgRc(t)}{AvgPr(t) + AvgRc(t)} \right\} \tag{5}$$

First, precision ($pr_i$) and recall ($rc_i$) are calculated according to Eqs (1) and (2). For every protein $i$ in the dataset, $f$ is a GO term label, $T_i$ is a set of true GO term labels, and $P_i(t)$ is a set of predicted GO term labels of threshold $t$. $I$ is an identity function which returns 1 for true and 0 for the false condition. Second, average precision ($AvgPr$) and average recall ($AvgRc$) are calculated as Eqs (3) and (4), where $m(t)$ is the number of proteins that contain at least one predicted label, and $n$ is the total number of proteins. The $F1_{max}$ is calculated as Eq (5) with the threshold $t$ in range of 0 to 1 and a step size of 0.1.

**Nested cross-validation.** We used five repetitions of nested 5x3-fold cross-validation (CV) to evaluate the model performance with the optimal parameter set. The schema of the nested CV is presented in S3 Fig. For every fold, a training set was separated and used to perform nested CV to find the best parameter set before validating with the test set. That meant we could test the model in each fold with the best parameter set without the leaking of information during the step of model training.

## Results

### Performance of PhotoModGO in comparison to the baseline methods

We applied PhotoModGO to predict proteins of functional subclasses in the photosynthetic system. Among the multi-label classifiers, RAkEL performed the best with an $F1_{max}$ value of 0.872, followed by the BR method with a value of 0.862 (Table 1). The performance between these two algorithms was not significantly different ($p$-value = 0.166) based on the Wilcoxon signed-rank test (S2 Table). LP performed the worst with an $F1_{max}$ value of 0.693, which was significantly different from the other two methods ($p$-value < 0.00001). Although the LP model showed the highest recall value, its precision was low compared to the others. This is consistent with a previous benchmark [31] showing that ensemble transformation methods generally perform better than simple transformation methods.

The same dataset was used to train two sequence-based models, DeepGoPlus and BLAST, for performance comparison. DeepGoPlus achieved an $F1_{max}$ value of 0.702, a precision value of 0.771, and a recall value of 0.693. The classical BLAST method had the worst performance, with $F1_{max}$, precision, and recall values of 0.603, 0.601, and 0.624, respectively. RAkEL and BR performed significantly ($p$< 0.00001) better than DeepGoPlus, but LP was not statistically different from DeepGoPlus ($p$ = 0.135) in this regard. The genome neighborhood-based models outperformed the sequence-based models on average.

**Table 1. Performance comparison of PhotoModGO and the baseline methods, DeepGOPlus and BLAST, using five replications of nested 5x3-fold cross validation.**

| Model | F1$_{max}$ (±SD) | Precision (±SD) | Recall (±SD) |
|---|---|---|---|
| PhotoModGO-RAkEL | 0.872 (0.035) | 0.926 (0.031) | 0.869 (0.042) |
| PhotoModGO-BR | 0.862 (0.029) | 0.902 (0.037) | 0.873 (0.042) |
| PhotoModGO-LP | 0.693 (0.046) | 0.665 (0.061) | 0.940 (0.021) |
| DeepGOPlus | 0.702 (0.032) | 0.771 (0.041) | 0.693 (0.048) |
| BLAST | 0.603 (0.049) | 0.601 (0.052) | 0.624 (0.050) |

Although PhotoModGORAkEL showed high predictive performance overall, we found that four GO terms could not be predicted by this model, as shown by the performance for each GO term in S1 Table. One of the reasons is that the number of instances is not enough for training the model. Moreover, the predictive performance of the phycobilisome term (GO:0030089; $F1_{max}$ = 0.548), which is marginally connected to the photosynthesis term, was not significantly different (one sample two-tailed t-test: $p < 0.01$) from the average performance per GO term ($F1_{max}$ = 0.563). This indicates that the model development protocol is not seriously biased toward the original 23 photosynthesis-specific GO terms. In addition, it suggests that this protocol can be applied to other photosynthesis-related functions and other systems.

## Sequence identity-independent performance of PhotoModGO

The more similar protein sequences are, the more they tend to have similar functions [32]. Therefore, we examined the role of sequence identity in our model performance. We created a new dataset, called the easy dataset, containing a sequence identity of ≤70%. PhotoModGO-R-AkEL, DeepGOPlus, and BLAST models were trained with this dataset and evaluated using five replications of nested 5x3-fold CV. We observed that DeepGoPlus and BLAST methods performed better on the easy dataset (identity ≤70%) compared to the dataset with identity≤50%, but we observed no difference in PhotoModGO performance (Fig 5), indicating that PhotmoModGO does not suffer from the situation where proteins with similar functions share low sequence identity scores.

DeepGoPlus achieved an $F1_{max}$ measure of 0.887 with the easy dataset, while BLAST method could achieve a value of 0.885. This better performance of sequence-based models indicates that they take advantage of the more similar sequence to predict the function. On the other hand, PhotoModGO could not take advantage of such sequence property, resulting in no significant difference between high and low sequence identity datasets in the prediction of performance. The reason is that PhotoModGO considers both sequence relationships and genome neighborhood profiles for the classification, i.e., weakly homologous sequences with the same genome neighborhood profile were generally classified under the same functional class.

## Functional prediction of novel photosynthetic proteins

Moreover, our new model revealed the potential to predict the function of novel photosynthetic genes / proteins. For example, we retrieved 17 recently identified photosynthetic proteins, which had never been deposited in the Uniprot database as of September 2016. As shown in Table 2, the functional annotations of many proteins from literature are poorly described, which makes it difficult to work with them. We could apply the new model to these proteins to specify functional classes related to photosynthesis. We found that many proteins, including RfpA, IflA and DpxA, were predicted to be related to PSII, while only IsiX was predicted to be related to PSI. Interestingly, gene *sll0272* has been documented as a homolog of *NdhV* in *Arabidopsis thaliana*. Its deletion reduces the activity of NADPH dehydrogenase (NDH-1)and NDH-1-dependent cyclic electron transport around photosystem I (NDH-CET), which plays a protective role against several stress conditions, e.g. high light intensity, salt and temperature [33–35]. It has been proposed that the sll0272 protein is a ferredoxin-binding domain of cyanobacterial NDH-1 localized in the thylakoid membrane [33]. Based on our model prediction, it was proposed that the product of this gene functions in the photosynthetic system as a regulator or stabilizer of PSII components. A few studies supporting this prediction showed that the absence of NDH-CET activity impairs PSII repairing process under heat stress

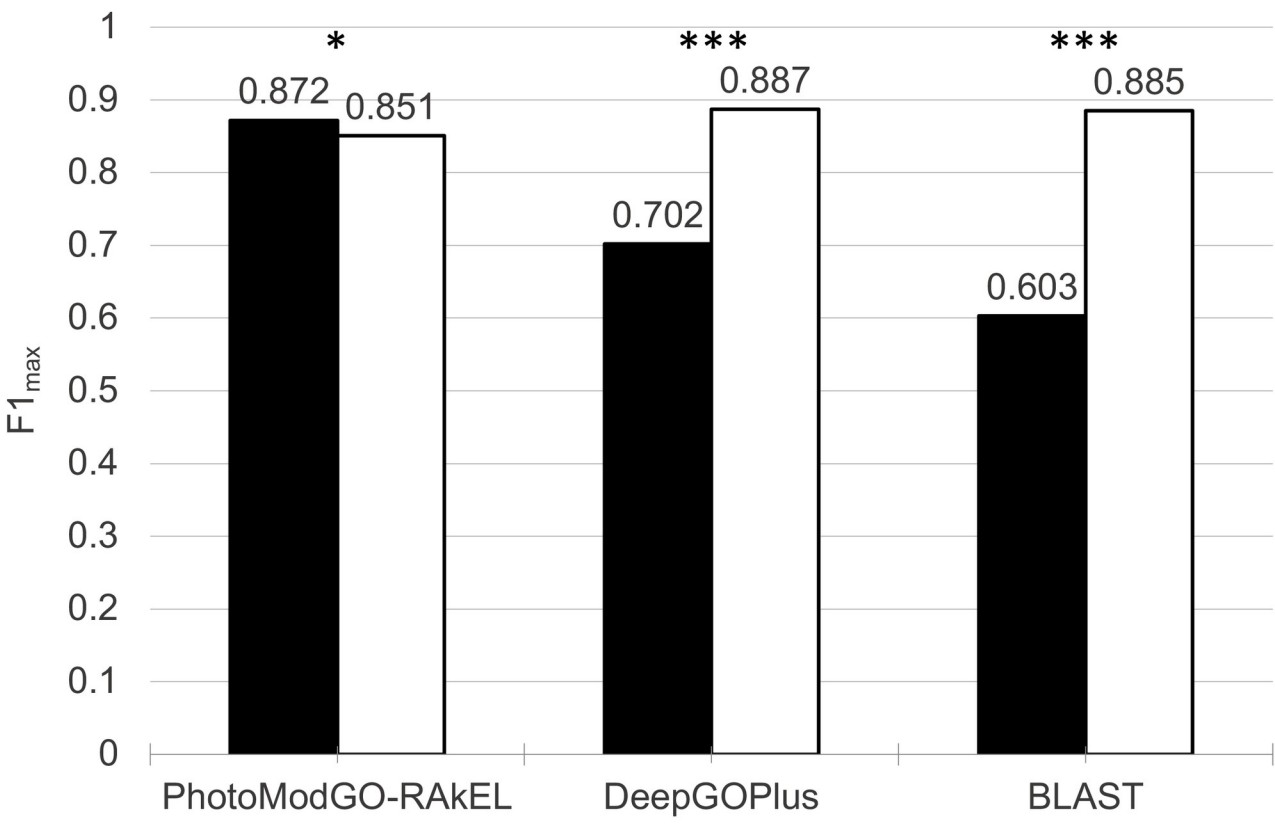

**Fig 5. PhotoModGO/multi-label classification performance of photosynthesis functions via 5 replications of nested 5x3 fold-cross validation.** The unique sequence dataset with ≤ 50% identity (diverse dataset) is represented by a black bar, while the dataset with ≤ 70% identity (easy dataset) is represented by a white bar. Asterisks indicate statistically significant differences, based on the Wilcoxon signed-rank test (*, $0.01 < p < 0.05$; ***, $p < 0.00001$).

conditions [36, 37]. Besides, three proteins, ApcD4, MpeZ and all4940, were predicted to be related to phycobilisomes as reported in the literature.

Surprisingly, we observed that the predicted GO terms from the DeepGoPlus model were sparse and not informative (S3 Table). For example, all of the query sequences were assigned by GO:0034357 (photosynthetic membrane) and GO:0009579 (thylakoid), while only ssl3829 and asr1131 were assigned by a more specific term, GO:0009523 (photosystem II). Moreover, only six of 17 query sequences could be assigned with photosynthetic GO terms by the SVMprot web server (S3 Table). These results indicated the advantage of PhotoModGO over the sequence-based approaches in the classification of photosynthetic subclasses.

## Functional prediction of unknown proteins in *Synechocystis* sp. PCC 6803

Identification of novel photosynthetic proteins might provide us a new route to improve photosynthetic efficiency. More than 50% (1,885 from 3672) of protein-coding genes in *Synechocystis* sp. PCC 6803, a model photosynthetic prokaryote, are unknown (data from http://genome.microbedb.jp/cyanobase, Sep 2018). Therefore, we applied two machine learning models, PhotoMod and PhotoModGO implemented in the PhotoModPlus web server, to

**Table 2. Functional prediction of novel photosynthetic proteins using PhotoModGO.**

| Protein name | Annotation from literature | Prediction (probability cut off ≥0.5) |
|---|---|---|
| RfpA | A regulator, which controls the expression of the Far-red light photoacclimation (FaRLiP) gene cluster [38] | GO:0009521 (Photosystem); GO:0009579 (thylakoid); GO:0009767 (photosynthetic electron transport); GO:0009772 (photosynthetic electron transport in photosystem II); GO:0019684 (photosynthesis light reactions); GO:0030096 (photosystem II (sensu Cyanobacteria)); GO:0034357 (photosynthetic membrane); GO:0045156 (electron transporter transferring electrons within the cyclic electron transport pathway of photosynthesis) |
| RfpB | A regulator, which controls the expression of the Far-red light photoacclimation (FaRLiP) gene cluster [38] | GO:0019685 (photosynthesis dark reactions) |
| IflA | A photoreceptor, which regulates cyanobacteriochrome (influenced by far-red light) [39] | GO:0009521 (Photosystem); GO:0009579 (thylakoid); GO:0009767 (photosynthetic electron transport); GO:0009772 (photosynthetic electron transport in photosystem II); GO:0019684 (photosynthesis light reactions); GO:0030096 (photosystem II (sensu Cyanobacteria)); GO:0034357 (photosynthetic membrane); GO:0045156 (electron transporter transferring electrons within the cyclic electron transport pathway of photosynthesis) |
| DpxA | A photoreceptor, which regulates cyanobacteriochrome by repressing phycoerythrin accumulation (in yellow light (570–590 nm) [40] | GO:0009521 (Photosystem); GO:0009579 (thylakoid); GO:0009767 (photosynthetic electron transport); GO:0009772 (photosynthetic electron transport in photosystem II); GO:0019684 (photosynthesis light reactions); GO:0030096 (photosystem II (sensu Cyanobacteria)); GO:0034357 (photosynthetic membrane); GO:0045156 (electron transporter transferring electrons within the cyclic electron transport pathway of photosynthesis) |
| FciA | A transcriptional regulator, which is responsible for tuning the phycourobilin: phycoerythrobilin ratio [41] | GO:0009521 (Photosystem); GO:0009579 (thylakoid); GO:0034357 (photosynthetic membrane) |
| FciB | A transcriptional regulator, which is responsible for tuning the phycourobilin: phycoerythrobilin ratio [41] | GO:0009521 (Photosystem); GO:0009579 (thylakoid); GO:0034357 (photosynthetic membrane) |
| IsiX | A specialized antenna protein, which functions under low irradiance or possibly far-red light (or both) conditions [42] | GO:0009521 (Photosystem); GO:0009522 (photosystem I); GO:0009579 (thylakoid); GO:0009767 (photosynthetic electron transport); GO:0019684 (photosynthesis light reactions); GO:0030094 (photosystem I (sensu Cyanobacteria)); GO:0034357 (photosynthetic membrane) |
| ApcD4 | A specialized antenna protein, which functions under low irradiance or possibly far-red light (or both) conditions [42] | GO:0009579 (thylakoid); GO:0030089 (phycobilisome); GO:0034357 (photosynthetic membrane) |
| MpeZ | A phycoerythrin-specific bilinlyase, which contributes to the type IV chromatic acclimation [41] | GO:0009579 (thylakoid); GO:0030089 (phycobilisome); GO:0034357 (photosynthetic membrane) |
| slr0151 | A protein involved in photosystem II assembly and repair [43] | GO:0009579 (thylakoid) |
| CyanoP | A protein involved in the early steps of photosystem II assembly in a cyanobacterium [44] | GO:0009579 (thylakoid) |
| slr1658 | A protein involved in the regulation of alternative electron flow in a cyanobacterium [45] | GO:0009579 (thylakoid); GO:0034357 (photosynthetic membrane) |
| sll0272 | A protein involved in the regulation of NDH-1 activity for efficient operation of cyclic electron flow around PS I and CO2 uptake, especially at high light [46] | GO:0009521 (Photosystem); GO:0009523 (photosystem II); GO:0009579 (thylakoid); GO:0010109 (regulation of photosynthesis); GO:0019684 (photosynthesis light reactions); GO:0034357 (photosynthetic membrane); GO:0042548 (regulation of photosynthesis light reaction); GO:0042549 (photosystem II stabilization) |
| ssl3829 | A cytoplasmic protein, which is involved in NDH-1 hydrophilic arm assembly [47] | GO:0009579 (thylakoid); GO:0034357 (photosynthetic membrane) |
| asr1131 | A Ca2+-binding protein, which influences photosynthetic electron transport [48] | GO:0009521 (Photosystem); GO:0009579 (thylakoid); GO:0034357 (photosynthetic membrane) |
| slr1188 | A protein, which triggers membrane fusion events in chloroplasts and cyanobacteria [49] | GO:0009579 (thylakoid); GO:0034357 (photosynthetic membrane) |
| all4940 | A protein, which can transfer carotenoid to helical carotenoid proteins [50] | GO:0009579 (thylakoid); GO:0030089 (phycobilisome); GO:0034357 (photosynthetic membrane) |

screen for unknown genes in the *Synechocystis* sp. PCC 6803 genome and identify potential novel photosynthetic genes. As a result, we could predict ~100 high confident photosynthetic gene candidates and their functions (as shown in S4 Table). Some of these interesting candidates with informative annotations are shown in Table 3.

**Table 3. Identification of potential photosynthetic genes in *Synechocystis* sp. PCC 6803 genome using PhotoModPlus.**

| Locus | GO prediction | Probability | Description |
|---|---|---|---|
| sll0249 | GO:0009521 | 1.0 | Photosystem |
| | GO:0009522 | 1.0 | photosystem I |
| | GO:0009579 | 1.0 | thylakoid |
| | GO:0034357 | 1.0 | photosynthetic membrane |
| sll0611 | GO:0009521 | 1.0 | Photosystem |
| | GO:0009523 | 1.0 | photosystem II |
| | GO:0009539 | 1.0 | photosystem II reaction center |
| | GO:0009579 | 1.0 | thylakoid |
| | GO:0030096 | 0.9 | photosystem II (sensu Cyanobacteria) |
| | GO:0034357 | 1.0 | photosynthetic membrane |
| slr0022 | GO:0009521 | 1.0 | Photosystem |
| | GO:0009523 | 1.0 | photosystem II |
| | GO:0009579 | 1.0 | thylakoid |
| | GO:0010109 | 1.0 | regulation of photosynthesis |
| | GO:0019684 | 1.0 | photosynthesis, light reactions |
| | GO:0034357 | 1.0 | photosynthetic membrane |
| | GO:0042548 | 1.0 | regulation of photosynthesis, light reaction |
| | GO:0042549 | 1.0 | photosystem II stabilization |
| slr2049 | GO:0009579 | 1.0 | thylakoid |
| | GO:0030089 | 1.0 | phycobilisome |
| | GO:0034357 | 1.0 | photosynthetic membrane |
| | GO:0019685 | 1.0 | photosynthesis, dark reactions |

As sll0249 is localized next to the *isiAB* operon, it was designated *isiC*. IsiA protein has been associated with PSI and forms a ring-like complex around the PSI reaction center under iron deficiency condition [51]. *isiC* was observed to be transcriptionally enhanced during iron deficiency with no functional designation [52]. Also, the *isiA* mutant was previously reported to be sensitive to high light [53], while the *isiC* mutant showed high sensitivity to oxidative stress [54]. It is known that oxygenic photosynthetic bacteria consume a large amount of iron for maintaining photosynthesis (e.g. a part of several iron-sulfur cluster-containing proteins). Thus, to minimize harmful radicals from oxidative interaction, iron accumulation in cells should be tightly regulated [55]. According to our prediction result and the pieces of evidence, we propose that sll0249 and IsiC play an important role in balancing PSI activity in response to oxidative stress.

Besides, the predicted result also demonstrated that sll0611 is associated with PSII. DNA microarrays of *Synechocystis* PCC6803 showed altered expression of the *sll0611* gene responding to the deletion of *lexA*, which is a key regulator of the SOS response induced by DNA damage [56]. A more comprehensive study achieved by RNA-seq analysis showed that deletion of the *lexA* gene resulted in altered expression of many genes, including those involved in photosynthesis, although no significant change in *sll0611* expression was observed in this study [57]. A further study focusing on sll0611 is required to clarify its link with photosynthesis. The slr0022 protein was previously identified in the core cyanobacterial clusters of orthologous groups of proteins (CyOGs) [11] and was predicted by the sequence similarity-based approach as Fe-S cluster protein. Our model was able to predict more specific functions of slr0022 related to photosynthesis. Additionally, our prediction result for slr2049 is consistent with a

recent publication [58] showing that slr2049 functions as lyase to catalyze phycobilin chromophores, which are a part of phycobiliproteins in phycobilisomes.

## Discussion

In our PhotoModPlus web server, we developed PhotoModGO, which integrates protein clustering, genome neighborhood profile generation, and multi-label classification algorithm to classify the photosynthesis subclasses. The model can assign photosynthetic functions to proteins with high accuracy, even though their sequences are highly diverse from the training sequences. This is an advantage over sequence-based approaches, including DeepGO and BLAST, which perform very well with the easy dataset but handles diverse datasets poorly. We demonstrated that the BLAST method performs efficiently when the sequences in the dataset are similar ($\leq$70% sequence identity). However, if the sequences in the dataset are more diverse ($\leq$50% sequence identity), the BLAST method suffers from a lack of information, resulting in low performance. Therefore, the machine-learning method plays an important role in this situation. We showed that the DeepGOPlus method, which combines sequence similarity-based prediction and the motif sequence-based neuron network model, performed better than the classical BLAST method with the diverse dataset. Moreover, the potential of PhotoModGO to uncover the photosynthetic function of novel photosynthetic proteins was demonstrated to provide more informative GO terms than that of the sequence-based approaches.

The sequence-based approaches are suited for use as a first tool for predicting protein functions are considering their speed, ease of use and performance efficiency. When higher precision is required or in the absence of similar sequences in the database, it is necessary to consider other suitable approaches carefully. We can use the information from other sources to predict functions, for example, protein-protein interaction, protein structure, and genomic contexts. The genomic context seems the cheaper and easier data option given the substantial reduction in the cost of genome sequencing. Two well-established genomic context-based methods are the phylogenetic profile [59, 60] and gene neighborhood [30]. The concept of both methods is similar, i.e., functionally linked proteins tend to co-occur in genomes [61]. However, gene neighborhood-based methods, particularly PhotoMod, focus on gene clusters and operons by observing only the surrounding region of the query gene. This generally can reduce noise or unrelated genes and provide high-quality functional relationships [15].

Additionally, by integrating the sequence similarity network and GNN [62], we developed PhotoModGNN to create and visualize GNN for inferring photosynthesis-related functions. Compared to the original GNN [62], PhotoModGNN was improved to circumvent complicated interpretations. One of the improvements is the use of Phylo scores as the indicative signal for detecting related functions from genome neighborhoods, instead of using the co-occurrence frequency, which might cause a bias toward dominant species in the dataset. In addition, enriched functions among genome neighborhoods are calculated and provided to users with the p-value to support users' decisions. One more feature of GNN is the ability to adjust the E-value threshold to separate protein families [62]. The range of the E-value threshold varies (ranging from 1E-8 to 1E-84) depending on the size and diversity of the dataset [62–64]. Therefore, we selected three E-value thresholds of 1E-10, 1E-50 and 1E-100 based on the minimum and maximum sequence similarity found in photosynthetic reaction center homologs. GNNs based on these three selected E-value thresholds are provided in PhotoModGNN to allow users to dynamically explore the genome neighborhood patterns from the different cutoffs. Therefore, PhotoModGNN can be used to explore functions of novel photosynthetic proteins other than the 24 subclasses that are predictable by PhotoModGO, as demonstrated.

Although PhotoModGO performs efficiently in predicting the photosynthetic subclasses, it is important to mention its limitations. Firstly, most of the photosynthetic-specific GO terms used in this study are in the biological process and cellular component categories (12 in the biological process, 10 in the cellular component, and 2 in molecular function). We showed that PhotoModGO generally outperformed the sequence-based approaches except for the molecular function category, where the sequence-based methods generally work well [65]. Thus, in the PhotoModPlus web server, we also enabled the prediction of functions based on the protein domain, via InterProScan [29], to improve the prediction quality. Secondly, gene neighborhood scoring is a time-consuming process. PhotoModGO calculates gene neighborhood conservation based on the phylogenetic tree, modified from Zheng et al. [66]. It requires a genome distance matrix based on shared gene contents between genomes to calculate the score. Our dataset contains 154 genomes, implying a huge amount of calculation (11,781 pairwise genome distance). Instead of using shared gene contents, one might estimate the distance between genomes using 16s rRNA, which can be obtained from public databases such as SILVA (https://www.arb-silva.de), to increase calculation speed and make the application easily expandable. Finally, our methods have been developed based on the concept of gene clusters and operons, which make it particularly applicable in only prokaryotic genomes, where gene clusters and operons commonly exist. Although we showed that genome neighborhood features could be used to classify photosynthetic proteins and their sub-functional classes, the framework can be applied to other systems. One of the most important systems on earth is the nitrogen assimilation and fixation system, which is crucial for carbon and nitrogen cycles and the global climate [67]. This complicated system is exclusively conducted by prokaryotes, and many related genes are organized as operons, which make it compatible with our framework.

## Conclusions

Among the diversity of available data types and computational algorithms for protein function prediction, the PhotoModPlus represents a new and unique method based on genome neighborhood analysis. The current version of the web server consists of i) PhotoMod: a genome neighborhood-based model to classify photosynthetic proteins [15], ii) PhotoModGO: a genome neighborhood-based multi-label model to classify sub-functional classes of photosynthesis, and iii) the PhotoModGNN platform for visualizing the genome neighborhood network to infer protein functions. Using genome neighborhood features with machine learning model PhotoModGO, we were able to increase the functional prediction performance by ~17–27% compared to other sequence-based approaches. Notably, the model can handle diverse sequence datasets well. The application of PhotoModPlus in the prediction of novel photosynthetic protein functions was demonstrated. We are assured that the user-friendly web interface of PhotoModPlus can increase its accessibility and usability for those interested in the discovery of photosynthetic genes / proteins.

### Implementation and availability

PhotoModGO is available as free software in Github at https://github.com/asangphukieo/PhotoModGO and in Dockerhub at https://hub.docker.com/r/asangphukieo/photomod2. The PhotoModPlus web server is available at bicep.kmutt.ac.th/photomod.

### Supporting information

**S1 Fig. Example of gene neighborhood and genome neighborhood network.** (A) Genes on the same strand are considered neighbors if they are within 250 bp intergenic distance or are overlapping. Additionally, the two clusters are merged into the same neighborhood gene

cluster if they are in a range of 200 to 1000 bp in the divergent direction, based on the operon interaction concept [68]. (B) Gene neighborhoods are called from all of the genomes that contain a homolog of the query sequence. (C) After applying protein clustering and calculating the Phylo score, a genome neighborhood network (GNN) can be constructed. The hexagonal node represents the query, while the circular node represents its genome neighbors. The label in each node represents a protein cluster ID. The size of the genome neighbor node varies according to its Phylo score. There are three built-in protein clustering cutoffs (E-value: 1E-10, 1E-50, and 1E-100) available on top of the network to define the level of homologous relationship between proteins.
(PDF)

**S2 Fig. Histogram of Phylo score distribution.** The Phylo scores were collected from the genome neighborhoods of the photosynthetic protein dataset (A) and nonphotosynthetic dataset (B). Note that the zero value of the Phylo score indicates a nonconserved genome neighborhood.
(PDF)

**S3 Fig. Demonstration of nested 5x3 fold-cross validation.** The outer fold is shown in the upper part of the figure, while the nested fold is shown in the lower part. For every outer fold, the training set (white color) is used to perform 3-fold cross-validation to find the best parameter set. The best parameter set is used to build a model again from the whole training set in the outer fold and tested with an independent test set (black color).
(PDF)

**S1 Table. List of 24 GO terms specific to the photosynthetic system and PhotoModGO prediction performance of each GO term.**
(PDF)

**S2 Table. P-value table for performance comparison of the multi-label photosynthetic function classification by the Wilcoxon signed-rank test.**
(PDF)

**S3 Table. GO term prediction result of 17 novel photosynthetic genes from the DeepGo-Plus model.**
(PDF)

**S4 Table. List of potential photosynthetic gene candidates predicted from the PhotoMod-Plus web server.** The unknown protein-coding genes from *Synechocystis*sp. PCC 6803 genome (1,885 genes) were used as input in PhotoModPlus prediction. BLAST best match and E-value of 1E-25 were used as BLAST parameters. The potential candidates were selected using the criteria: i) predicted as a positive class with more than 80% probability from the PhotoMod binary model and ii) containing at least one predicted GO term with more than 50% probability from the PhotoModGO multi-label model.
(PDF)

## Acknowledgments

The authors thank Dr. Weerayuth Kittichotirat and Dr. Sawannee Sutheeworapong for insightful discussions and suggestions. We also thank Bioinformatics and Systems Biology Program and King Mongkut's University of Technology Thonburi for the research support, materials and equipment. Furthermore, we thank Mr. Oscar Nnaemeka from the School of Bioresources and Technology, KMUTT, for his editing and proofreading of the manuscript.

## Author Contributions

**Conceptualization:** Apiwat Sangphukieo, Teeraphan Laomettachit, Marasri Ruengjitchatchawalya.

**Data curation:** Apiwat Sangphukieo.

**Formal analysis:** Apiwat Sangphukieo.

**Funding acquisition:** Teeraphan Laomettachit, Marasri Ruengjitchatchawalya.

**Investigation:** Apiwat Sangphukieo.

**Methodology:** Apiwat Sangphukieo.

**Project administration:** Teeraphan Laomettachit, Marasri Ruengjitchatchawalya.

**Resources:** Apiwat Sangphukieo, Teeraphan Laomettachit, Marasri Ruengjitchatchawalya.

**Software:** Apiwat Sangphukieo.

**Supervision:** Teeraphan Laomettachit, Marasri Ruengjitchatchawalya.

**Validation:** Apiwat Sangphukieo.

**Visualization:** Apiwat Sangphukieo.

**Writing – original draft:** Apiwat Sangphukieo.

**Writing – review & editing:** Apiwat Sangphukieo, Teeraphan Laomettachit, Marasri Ruengjitchatchawalya.

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
