## [Decision Letter · Decision Letter 0]

4 Jan 2021

PONE-D-20-18308

PhotoModPlus: A web server for photosynthetic protein prediction from genome neighborhood features

PLOS ONE

Dear Dr. Ruengjitchatchawalya,

Thank you for submitting your manuscript to PLOS ONE. After careful consideration, we feel that it has merit but does not fully meet PLOS ONE’s publication criteria as it currently stands. Therefore, we invite you to submit a revised version of the manuscript that addresses the points raised during the review process.

The reviewers have raised some critical concerns that should be addressed.

1) The structure and flow of the manuscript should be re-organised.

2) The method should be compared with other existing methods and the reasons behind the outperformance should be justified.

3) More technical details should be added to improve the clarity and readability of the manuscript.

We look forward to receiving your revised manuscript.

Kind regards,

Dapeng Wang

Academic Editor

PLOS ONE

"We also thank Bioinformatics and Systems Biology Program and King Mongkut's University of Technology Thonburi for the research support, materials and equipment."

"This work was partly supported by Petchra Pra JomKlao Doctoral Scholarship (No: 13/2558) from King Mongkut's University of Technology Thonburi and a research grant (NRMJ: 2559A30602134#60000108) from the National Research Council of Thailand (http://en.nrct.go.th). The funders had no role in the study design, data collection and analysis, decision to publish, and preparation of the manuscript."

4. We suggest you thoroughly copyedit your manuscript for language usage, spelling, and grammar. If you do not know anyone who can help you do this, you may wish to consider employing a professional scientific editing service.  

Reviewers' comments:

Reviewer's Responses to Questions

**Comments to the Author**

1. Is the manuscript technically sound, and do the data support the conclusions?

Reviewer #1: Yes

Reviewer #2: Yes

2. Has the statistical analysis been performed appropriately and rigorously? 

Reviewer #1: Yes

Reviewer #2: Yes

3. Have the authors made all data underlying the findings in their manuscript fully available?

Reviewer #1: Yes

Reviewer #2: Yes

4. Is the manuscript presented in an intelligible fashion and written in standard English?

Reviewer #1: Yes

Reviewer #2: Yes

5. Review Comments to the Author

Reviewer #1: The paper presents a web server, namely PhotoModPlus, that offers three main functionalities; (1) constructing a machine learning model using the previously published PhotoMod algorithm, (2) predicting sub-functional classes of a photosynthetic function using a novel machine learning algorithm called PhotoModGO, and (3) generating and visualizing a genome neighborhood network (GNN) using a module called PhotoModGNN.

After providing a quick context on the importance of identifying genes and proteins relevant to photosynthesis, the manuscript offers an overview of the existing efforts and challenges in correctly classifying photosynthetic proteins; the main challenge being the impact of the increased diversity / evolutionary distance to the algorithms that rely on sequence features for classification (two of these methods, BLAST and DeepGOPlus, where used as a comparison). The article continues with an overview of the different methods and their respective implementations, and concludes with some results on selected species.

Overall it's an interesting work, but there are a few points that may be worth re-assessing

- The main issue is that the manuscript has a rather convoluted structure, thus hindering the readers' understanding, as it requires multiple passes in order to get a clear picture. One suggestion would be to include in the "Methods" section both of the different methods described ("PhotoModGO development" and "PhotoModGNN development") including the data collection sections, and add a new subsection on "Implementation" that could include the "PhotoModPlus web server implementation" and possibly also the "PhotoModPlus web server demonstration". The "Results" section could then incorporate the "Baseline comparison methods" and "Evaluation" as the initial subsections, so that there is a clear connection to the corresponding results. However, this is only a suggestion; any coherent re-structuring, that is better aligned to the message to be conveyed, would work equally well.

- The definition of genome neighborhood profiles seems quite similar to that of the phylogenetic profiles (https://doi.org/10.1073/pnas.96.8.4285) - or rather generalized version of them (https://doi.org/10.1371/journal.pone.0052854) - with a main challenge being that they require sufficient data (https://doi.org/10.1371/journal.pone.0114701), as well as they are rather sensitive to noise. If there is indeed a similarity, it would be crucial to establish what the effect of these aspects might be to the outcomes of the PhotoModGO algorithm. If not, it would be useful to clarify what the difference might be, especially given the fact that the underlying technique in both of them is sequence-based similarity using BLAST.

- The distinction between the original PhotoMod and the newly proposed PhotoModGO algorithm is not very clear. It is understood that PhotoMod can construct a binary (single-class) model, whereas PhotoModGO is more of a workflow that "wraps" around RelieF-ML and Random Forests in order to produce prediction models of the various photosynthesis-related GO terms. If that is the case, it should be clearly stated as such, in order to avoid confusion between the two different stages. If not, the corresponding section should be rephrased in order to clarify the message. In any case, a clear outline of the PhotoModGO method (ideally in the form of a pseudo-code) would be useful to provide additional clarity. Moreover, it should be clarified whether PhotoModGO can be used in place of PhotoMod (for the binary classification) as well.

- An additional point here is to clarify which types of Gene Ontology terms were used in the retrieval process; given that there are three distinct term trees (Molecular Function, Biological Process, and Cellular Component), it should be clarified whether the 61 photosynthetic specific GO terms (as performed in 2012) are part of the same tree, or can be assessed independently via a different mechanism. Moreover, given that the main focus of this effort is to outperform sequence-based algorithms, it may be useful to explore terms that are marginally connected to photosynthesis - as the photosynthesis-specific terms may introduce an additional bias to the training of the model (taking the whole process, from PhotoMod to PhotoModGO / PhotoModGNN into consideration).

- A final point is related to the discussion; it's absolutely clear that PhotoModPlus outperforms the two selected sequenced-based algorithms (diamond BLAST and DeepGOPlus). Would it be possible to re-use/adapt PhotoModPlus to work in a different context than photosynthesis - e.g. used to classify nitrate-reducing bacteria, and consequently further classify them across the different relevant GO terms. A short comment to this end, as well as some insights on the technical complexity required, would be useful.

Minor points:

- All figures are of extremely low quality, making some very hard to understand (especially Figure 2, 4 and 5).

- Why are there two servers / URLs listed (i.e. http://bicep.kmutt.ac.th/photomod and http://bicep2.kmutt.ac.th/photomod)? If there is a clear distinction (i.e. different underlying resources, available services etc), it should be noted as such - or at least clarified in order to avoid confusion.

- The sharing of the code and data by the authors is exemplary. However, all repositories should clearly include an appropriate License as well as a citation/acknowledgement file, in order to facilitate the (re-)use of the code by the wider community.

Reviewer #2: This MS developed a new method for predicting photosynthetic proteins. It was properly benchmarked and showed better performance than sequence based methods. The only problem is that at the time

of this review, the server can not be accessed.

6. PLOS authors have the option to publish the peer review history of their article (what does this mean?). If published, this will include your full peer review and any attached files.

Reviewer #1: **Yes: **Fotis E. Psomopoulos

Reviewer #2: No

---

## [Author Response · Author response to Decision Letter 0]

10 Feb 2021

Response to Reviewers

PONE-D-20-18308 

PhotoModPlus: A web server for photosynthetic protein prediction from genome neighborhood features

PLOS ONE Academic Editor’s decision/comment/suggestion:

After careful consideration, we feel that it has merit but does not fully meet PLOS ONE’s publication criteria as it currently stands. Therefore, we invite you to submit a revised version of the manuscript that addresses the points raised during the review process.

The reviewers have raised some critical concerns that should be addressed.

1) The structure and flow of the manuscript should be re-organised.

2) The method should be compared with other existing methods and the reasons behind the outperformance should be justified.

3) More technical details should be added to improve the clarity and readability of the manuscript.

Response: We have answered and changed regarding the comment/ suggestion/ critical concerns as the follows. 

Reviewers' comments:

Reviewer #1: The paper presents a web server, namely PhotoModPlus, that offers three main functionalities; (1) constructing a machine learning model using the previously published PhotoMod algorithm, (2) predicting sub-functional classes of a photosynthetic function using a novel machine learning algorithm called PhotoModGO, and (3) generating and visualizing a genome neighborhood network (GNN) using a module called PhotoModGNN.

After providing a quick context on the importance of identifying genes and proteins relevant to photosynthesis, the manuscript offers an overview of the existing efforts and challenges in correctly classifying photosynthetic proteins; the main challenge being the impact of the increased diversity / evolutionary distance to the algorithms that rely on sequence features for classification (two of these methods, BLAST and DeepGOPlus, where used as a comparison). The article continues with an overview of the different methods and their respective implementations and concludes with some results on selected species.

Overall it's an interesting work, but there are a few points that may be worth re-assessing

- The main issue is that the manuscript has a rather convoluted structure, thus hindering the readers' understanding, as it requires multiple passes in order to get a clear picture. One suggestion would be to include in the "Methods" section both of the different methods described ("PhotoModGO development" and "PhotoModGNN development") including the data collection sections, and add a new subsection on "Implementation" that could include the "PhotoModPlus web server implementation" and possibly also the "PhotoModPlus web server demonstration". The "Results" section could then incorporate the "Baseline comparison methods" and "Evaluation" as the initial subsections, so that there is a clear connection to the corresponding results. However, this is only a suggestion; any coherent re-structuring, that is better aligned to the message to be conveyed, would work equally well.

Response: The change was made as suggested by the reviewer. We have revised structure of the manuscript in the Methods and Results sections as the follows, as well as several revised sentences in parts, including discussion and conclusion, for a better conveyed message.

Methods

1st submitted version

• PhotoModPlus web server implementation

• Dataset collection

-Training and test datasets

- Novel photosynthetic protein dataset

• PhotoModGO development

- Nested cross-validation

- Evaluation metric

o Baseline comparison methods: PhotoModGO v.s. DeepGO Plus & Blast

• PhotoModGNN development

Revised version

• Dataset collection

-Training and test datasets

- Novel photosynthetic protein dataset

• PhotoModGO development

• PhotoModGNN development

• PhotoModPlus web server implementation and demonstration

- Photosynthetic function prediction by PhotoModGO

- Functional guidance by PhotoModGNN

• Baseline comparison methods and evaluation: PhotoModGO v.s. DeepGO Plus & Blast

- Evaluation metric

- Nested cross-validation

Results

1st submitted version

• Performance of PhotoModGO in multi-label classification

• Sequence identity-independent performance of PhotoModGO

• Functional prediction of novel photosynthetic proteins

• Functional prediction of unknown proteins in Synechocystis sp. PCC 6803

• PhotoModPlus web server demonstration

- Photosynthetic function prediction by the machine learning model

- Functional guidance by a genome neighborhood network

Revised version

• Performance of PhotoModGO in comparison to the baseline methods

• Sequence identity-independent performance of PhotoModGO

• Functional prediction of novel photosynthetic proteins

• Functional prediction of unknown proteins in Synechocystis sp. PCC 6803

- The definition of genome neighborhood profiles seems quite similar to that of the phylogenetic profiles (https://doi.org/10.1073/pnas.96.8.4285) - or rather generalized version of them (https://doi.org/10.1371/journal.pone.0052854) - with a main challenge being that they require sufficient data (https://doi.org/10.1371/journal.pone.0114701), as well as they are rather sensitive to noise. If there is indeed a similarity, it would be crucial to establish what the effect of these aspects might be to the outcomes of the PhotoModGO algorithm. If not, it would be useful to clarify what the difference might be, especially given the fact that the underlying technique in both of them is sequence-based similarity using BLAST. 

Response: Phylogenetic profile-based methods [1,2] and the gene neighborhood-based methods [3] predict gene function by considering co-occurrence of genes in different genomes, however, the major difference is that the latter decreases noise or un-related genes by observing only surrounded region of the query gene. This concept is closer to the group of functional-linked genes or operon, which is commonly found in bacteria. Moreover, in PhotoModGO we applied phylogeny-based score (Phylo score) modified from Zheng et al. [4] to measure gene neighborhood conservation without bias towards predominant phylum. We used three stringency criteria (1E-10, 1E-50 and 1E-100) for detecting homologous sequence in genomes, which is different from other approaches. The original concept is from the sequence similarity network [5], which employs multiple e-value thresholds to differentiate protein function. This feature was included in our algorithm, which was demonstrated to improve prediction performance in our previous publication [6]. Accordingly, we have rephrased Line: 456-464.

References: 

[1] Pellegrini M, Marcotte EM, Thompson MJ, Eisenberg D, Yeates TO. Assigning protein functions by comparative genome analysis: protein phylogenetic profiles. Proc Natl Acad Sci U S A. 1999;96(8):4285-8.

[2] Psomopoulos FE, Mitkas PA, Ouzounis CA. Detection of genomic idiosyncrasies using fuzzy phylogenetic profiles. PLoS One. 2013;8(1):e52854.

[3] Zhao S, Sakai A, Zhang X, Vetting MW, Kumar R, Hillerich B, et al. Prediction and characterization of enzymatic activities guided by sequence similarity and genome neighborhood networks. Elife. 2014;3:e03275.

[4] Zheng Y, Anton BP, Roberts RJ, Kasif S. Phylogenetic detection of conserved gene clusters in microbial genomes. BMC Bioinform. 2005;6:243 

[5] Atkinson HJ, Morris JH, Ferrin TE, Babbitt PC (2009) Using Sequence Similarity Networks for Visualization of Relationships Across Diverse Protein Superfamilies. PLOS ONE 4(2): e4345. https://doi.org/10.1371/journal.pone.0004345

[6] Sangphukieo, A., Laomettachit, T. & Ruengjitchatchawalya, M. Photosynthetic protein classification using genome neighborhood-based machine learning feature. Sci Rep 10, 7108 (2020). https://doi.org/10.1038/s41598-020-64053-w

- The distinction between the original PhotoMod and the newly proposed PhotoModGO algorithm is not very clear. It is understood that PhotoMod can construct a binary (single-class) model, whereas PhotoModGO is more of a workflow that "wraps" around RelieF-ML and Random Forests in order to produce prediction models of the various photosynthesis-related GO terms. If that is the case, it should be clearly stated as such, in order to avoid confusion between the two different stages. If not, the corresponding section should be rephrased in order to clarify the message. In any case, a clear outline of the PhotoModGO method (ideally in the form of a pseudo-code) would be useful to provide additional clarity. Moreover, it should be clarified whether PhotoModGO can be used in place of PhotoMod (for the binary classification) as well.

Response: PhotoModGO is not for replacing PhotoMod, but it is used as post processing step after obtaining the list of photosynthetic protein candidates from PhotoMod, as shown in Fig 1.

Our PhotoMod (binary classifier) was previously developed to classify the photosynthetic protein and non-photosynthetic protein, whereas the PhotoModGO (multi-label classifier) is further developed to classify photosynthetic function of the protein based on a genome neighborhood feature as stepwise outline shown in Fig 2. The workflow shows the sharing of data preparation and preprocessing between PhotoMod and PhotoModGO in the web server. Additionally, the PhotoModGO includes multi-label feature selection, RelieF-ML method, to determine the optimal set of features by removing redundant and irrelevant features to increase learning speed and model performance; and also employs RAndom k labEL sets (RAkEL), a problem transformation algorithm, to transform the multi-label dataset into a multi-class dataset before training random forest algorithm to generate model. 

We have rephrased the sentences in Line: 49-58; 84-88; 128-135. 

We have also modified Fig 2 and rephrased the figure legend (Line: 195-198), in addition to the sentences in Line: 128-135 to clarify the difference between PhotoMod and PhotoModGO.

- An additional point here is to clarify which types of Gene Ontology terms were used in the retrieval process; given that there are three distinct term trees (Molecular Function, Biological Process, and Cellular Component), it should be clarified whether the 61 photosynthetic specific GO terms (as performed in 2012) are part of the same tree, or can be assessed independently via a different mechanism. Moreover, given that the main focus of this effort is to outperform sequence-based algorithms, it may be useful to explore terms that are marginally connected to photosynthesis - as the photosynthesis-specific terms may introduce an of the model (taking the whole process, from PhotoMod to PhotoModGO / PhotoModGNN into consideration).

Response: As suggested by the reviewer, we clarify that the 61 photosynthesis-specific GO terms are retrieved from all child terms (“is_a” and “part_of” relationships) of photosynthesis term (GO:0015979) that are not part of other nonphotosynthetic functions in biological processes. Therefore, they can be considered as a part of the same tree. We provided retrieval process and more information of each GO term used in our model including main category (Molecular Function, Biological Process, and Cellular Component) to supplementary S1 Table. Moreover, we provided the level of GO term in the tree (shortest and longest path from the main category term) to show information content of each GO term. We demonstrated an example of useful exploring Phycobilisome (GO:0030089), which is in Cellular component category and is not directly linked to photosynthesis term in Biological process category. We measured the prediction performance of each GO term and found that of the phycobilisome (F1max=0.548) was not significantly different (one sample two tailed t-test: p < 0.01) from the average performance per GO term (F1max=0.563) indicating there is no serious bias of model development protocol to the list of original 23 photosynthesis-specific GO terms. In addition, it suggests that this protocol can be applied to other photosynthesis-related functions and other systems. 

Appropriately, we have rephrased the sentences (Line: 135-144; 331-336) and added information on GO term retrieval process in supplementary S1 Table.

- A final point is related to the discussion; it's absolutely clear that PhotoModPlus outperforms the two selected sequenced-based algorithms (diamond BLAST and DeepGOPlus). Would it be possible to re-use/adapt PhotoModPlus to work in a different context than photosynthesis - e.g. used to classify nitrate-reducing bacteria, and consequently further classify them across the different relevant GO terms. A short comment to this end, as well as some insights on the technical complexity required, would be useful.

Response: Definitely, our platform would be effectively applicable to other systems whose genes are organized into clusters or operons on the genomes. Possible examples would be nitrogen assimilation and fixation pathways, which is one of the most important systems on earth crucial for carbon and nitrogen cycles. This complicated system is exclusively conducted by prokaryotes, and many related genes are organized as operons, which make it compatible with our framework. However, the limitation of our method is computational complexity of Phylo score or gene neighborhood conservation score. To calculate this score, it requires the genome distance matrix based on shared gene content between genomes, modified from Zheng et al. [1]. For example, our dataset contains 154 genomes implying a large amout of calculation of 11,781 pairwise genome distance which consumes large computational resource. However, the calculation speed can be increased by estimating the distance between genomes by using 16s rRNA obtained from public databases, such as SILVA (https://www.arb-silva.dc). We have appropriately rephrased the sentences (Line: 489-504).

[1] Zheng Y, Anton BP, Roberts RJ, Kasif S. Phylogenetic detection of conserved gene clusters in microbial genomes. BMC Bioinform. 2005;6:243

Minor points:

- All figures are of extremely low quality, making some very hard to understand (especially Figure 2, 4 and 5).

Response: Actually, we have submitted all the figures with high resolution. We have re-checked figure quality in https://pacev2.apexcovantage.com and there is no any issue found. Probably, the problem might occur during PDF conversion step in PlosOne server. We will notify this issue to editors. 

- Why are there two servers / URLs listed

 (i.e. http://bicep.kmutt.ac.th/photomod and http://bicep2.kmutt.ac.th/photomod)? If there is a clear distinction (i.e. different underlying resources, available services etc), it should be noted as such - or at least clarified in order to avoid confusion.

Response: There is no different use of both URLs. Both URLs point to the same computer server, but using different path to avoid internet traffic issue. Currently, we used

http://bicep.kmutt.ac.th/photomod as main URL. 

We removed http://bicep2.kmutt.ac.th/photomod.

- The sharing of the code and data by the authors is exemplary. However, all repositories should clearly include an appropriate License as well as a citation/acknowledgement file, in order to facilitate the (re-)use of the code by the wider community.

Response: As suggested by the reviewer, we have added License and citations in our data and code. 

Reviewer #2: This MS developed a new method for predicting photosynthetic proteins. It was properly benchmarked and showed better performance than sequence based methods. The only problem is that at the time of this review, the server can not be accessed.

Response: We apologies for our technical issue. We try our best to maintain our server. One of the issues is domain traffic problem, which we resolved by generating a new optional URL (http://bicep2.kmutt.ac.th/photomod) using different path to go to our server.

---

## [Decision Letter · Decision Letter 1]

4 Mar 2021

PhotoModPlus: A web server for photosynthetic protein prediction from genome neighborhood features

PONE-D-20-18308R1

Dear Dr. Ruengjitchatchawalya,

We’re pleased to inform you that your manuscript has been judged scientifically suitable for publication and will be formally accepted for publication once it meets all outstanding technical requirements.

Kind regards,

Dapeng Wang

Academic Editor

PLOS ONE

Reviewers' comments:

Reviewer's Responses to Questions

**Comments to the Author**

1. If the authors have adequately addressed your comments raised in a previous round of review and you feel that this manuscript is now acceptable for publication, you may indicate that here to bypass the “Comments to the Author” section, enter your conflict of interest statement in the “Confidential to Editor” section, and submit your "Accept" recommendation.

Reviewer #1: All comments have been addressed

Reviewer #2: (No Response)

2. Is the manuscript technically sound, and do the data support the conclusions?

Reviewer #1: Yes

Reviewer #2: (No Response)

3. Has the statistical analysis been performed appropriately and rigorously? 

Reviewer #1: Yes

Reviewer #2: (No Response)

4. Have the authors made all data underlying the findings in their manuscript fully available?

Reviewer #1: Yes

Reviewer #2: (No Response)

5. Is the manuscript presented in an intelligible fashion and written in standard English?

Reviewer #1: Yes

Reviewer #2: (No Response)

6. Review Comments to the Author

Reviewer #1: The authors have addressed all comments raised in the first round of reviewing, and I have no further concerns about this manuscript. I find it acceptable.

Reviewer #2: (No Response)

7. PLOS authors have the option to publish the peer review history of their article (what does this mean?). If published, this will include your full peer review and any attached files.

Reviewer #1: **Yes: **Fotis E. Psomopoulos

Reviewer #2: No

---

## [Editor Report · Acceptance letter]

8 Mar 2021

PONE-D-20-18308R1 

PhotoModPlus: A web server for photosynthetic protein prediction from genome neighborhood features 

Dear Dr. Ruengjitchatchawalya:

I'm pleased to inform you that your manuscript has been deemed suitable for publication in PLOS ONE. Congratulations! Your manuscript is now with our production department. 

Kind regards, 

on behalf of

Dr. Dapeng Wang 

Academic Editor

PLOS ONE